# Systematic and meta-analysis of factors associated with preeclampsia and eclampsia in sub-Saharan Africa

**Maereg Wagnew Meazaw**[1,2]*, **Catherine Chojenta**[2], **Muluken Dessalegn Muluneh**[3,4], **Deborah Loxton**[2]

**1** Federal Ministry of Health, Addis Ababa, Ethiopia, **2** Research Centre for Generational Health and Ageing, School of Medicine and Public Health, Faculty of Health and Medicine, University of Newcastle, Newcastle, Australia, **3** School of Nursing and Midwifery, Western Sydney University, Parramatta, Australia, **4** Amref Health Africa in Ethiopia, Addis Ababa, Ethiopia

* mulemar12@gmail.com

**Data Availability Statement:** All relevant data are within the manuscript and its Supporting Information files.

## Abstract

### Background

Preeclampsia and eclampsia are common complications of pregnancy globally, including sub-Saharan African (SSA) countries. Although it has a high burden on maternal and neonatal mortality and morbidity, evidence on the risk of the problem is limited. Therefore, the aim of this review was to examine the factors associated with preeclampsia and eclampsia among mothers in SSA countries.

### Methods

We searched article from SSA countries using electronic database MEDLINE, EMBASE, PubMed, CINAHL published in English from January 2000 to May 2020. Two reviewers independently screened, extracted and assessed the quality of the articles. Both random and fixed effect model were used for analysis. Heterogeneity of the studies and publication bias were checked. STATA 16 used for analysis.

### Results

Fifty-one studies met the inclusion criteria and included in this review. The following factors were identified through meta-analysis: being primiparous (OR: 2.52; 95% CI:1.19, 3.86), previous history of maternal preeclampsia/eclampsia (OR:5.6; 95% CI:1.82, 9.28), family history of preeclampsia/eclampsia (OR:1.68; 95% CI:1.26, 2.11), high maternal body mass index (OR: 1.69; 95% CI:1.17, 2.21), chronic hypertension (OR: 2.52; 95% CI:1.29, 3.74), anaemia during pregnancy (OR: 3.22; 95% CI:2.70, 3.75) and lack of antenatal care visits (OR: 2.71; 95% CI:1.45, 3.96). There was inconclusive evidence for a relationship with a number of other factors, such as nutrition and related factors, antenatal care visits, birth spacing, and other factors due to few studies found in our review.

**Funding:** This research received no external funding.

**Competing interests:** The authors declare no conflicts of interest.

## Conclusions

The risk of preeclampsia and eclampsia is worse among women who have a history of pre-eclampsia/eclampsia (either themselves or family members), primiparous, obesity and over-weight, living with chronic disease, having anaemia during pregnancy and absence from ANC visits. Therefore, investment must be made in women's health needs to reduce the problem and health service providers need to give due attention to high-risk women.

## Introduction

Globally, preeclampsia and eclampsia contribute to the death of a pregnant woman every three minutes [1]. Preeclampsia and eclampsia remain among the top causes of maternal and foetal morbidity and mortality worldwide [2]. According to the American College of Obstetrics and Gynaecology (ACOG) definition, preeclampsia is defined as the presence of hypertension and proteinuria ($\geq$140mmHg/90mmHg) or in absence of proteinuria, new-onset hypertension with the new onset of any of the following: thrombocytopenia, renal insufficiency, impaired liver function, pulmonary edema, and unexplained new-onset headache unresponsive to medication or visual symptoms [3]. The hypertension appear after 20 weeks gestation in previously normotensive women on at least two occasions more than four hours apart and resolving completely by the 6th postpartum week [3]. If left untreated, preeclampsia can develop into eclampsia, which is the occurrence of generalized seizures with preeclampsia, if the tonic-clonic seizures are not attributable to other causes (e.g., epilepsy or brain tumour) [4].

Worldwide, the incidence of preeclampsia and eclampsia varies country to country, and it is estimated that it affects between 2% and 10% of pregnancies every year [2]. A systematic review from 40 countries that included nearly 39 million women estimated that 4.6% and 1.4% of all deliveries were complicated by preeclampsia and eclampsia, respectively, with a wide variation across regions [5]. According to the World Health Organization (WHO) estimates, the incidence of preeclampsia in developing countries (2.8% of live births) is seven times higher than in developed countries (0.4% of live births) [2,6]. Eclampsia also increases the risk of maternal death both in developed (0.5–1.8%) and in developing countries (15%) [7]. Evidence shows that in SSA countries preeclampsia and eclampsia are among the top five leading causes of morbidity and mortality of women and babies [8]. Because of this and other factors, SSA countries experience the highest maternal and newborn mortality [9].

Many researchers have studied the underlining causes of preeclampsia in different settings. However, the exact aetiology of the condition is unknown [10].

Different studies conducted in different parts of the world have reported a range of risk factors, although results were inconclusive due to variations among populations and ethno-geographic groups [11]. In addition, there are inconsistencies in reporting risk factors across the literature. Moreover, most of the evidence reported about risk factors of preeclampsia and eclampsia were from developed countries. Given the ethno-geographic differences in risk factors and the high rate of maternal mortality in SSA, there is a need for understanding risk factors for preeclampsia and eclampsia at the regional level. However, studies conducted on risk factors for preeclampsia and eclampsia across different SSA countries have been at the facility-based level and involved small, small sample sizes [12–14].

To date there has not been a study conducted that collectively and systematically analyses risk factors of preeclampsia and eclampsia in SSA. Therefore, this study aimed to systematically explore the pooled risk factors for preeclampsia/eclampsia in SSA countries. This fill the

gap in evidence on the strength of maternal and newborn health programs, policies and strategies that aim to reduce maternal and neonatal mortality and morbidity due to preeclampsia and eclampsia.

## Method

### Information source

A search of electronic databases including MEDLINE, EMBASE, Maternity and Infant Care and CINAHL was undertaken. Prior to starting this systematic review, we ensured that the research question did not appear in any existing review using the Cochrane, Health Services Research Projects in Progress (HSRProj) and Prospero International Prospective Register of Systematic Reviews (PROSPERO) database registries. Librarians were also consulted during the design of our search strategy and the search for articles from the above databases.

### Inclusion and exclusion criteria

Articles that were published on SSA data between January 2000 and May 2020 were included. The year 2000 was set as the starting point because many maternal health programs were introduced as a result of the Millennium Development Goals. Peer reviewed studies on preeclampsia and eclampsia that were conducted in SSA countries and were published in English were retrieved and included in the systematic review and meta-analysis. Articles published before January 2000, governmental and non-governmental reports, letters to the editor, opinion articles, and studies other than those that involved a case with preeclampsia or eclampsia were excluded.

### Search strategy

This study followed the PRISMA (Preferred Reporting Items for Systematic Reviews and Meta-Analyses) guideline [15]. A systematic search, based on the selection criteria and combining Medical Subject Headings (MeSH) terms, was developed using the above databases. The search strategy was limited to English and published articles since 2000. The search terms/keywords first used in MEDLINE were adapted to the other databases that are mentioned above. In addition, the reference section from identified articles was reviewed for additional articles. The following search terms and words were used to perform the search review.

hypertension, pregnancy-induced/ or eclampsia/ or hellp syndrome/ or pre-eclampsia/, eclamp*.mp., (preeclamp* or pre-eclamp*).mp., (hypertens* adj3 (preg* or gestat*)).mp.

study setting: (Subsahara* or Sub-Sahara* or Ethiopia* or Kenya* or Somalia* or Sudan* or Tanzania* or Uganda* or Eriteria* or Burundi* or Angola* or Benin* or Botswana* or Burkina Faso* or Cameroon* or Cape Verde* or Central African Republic* or Chad* or Comoros* or Congo * or Cote d'Ivoire* or Djibouti* or Equatorial Guinea* or Gabon* or Gambia * or Ghana* or Guinea * or Lesotho* or Liberia * or Madagascar* or Malawi * or Mali* or Mauritania* or Mauritius* or Mozambique* or Namibia * or Niger* or Nigeria* or Rwanda* or Sao Tome or Principe* or Senegal* or Seychelles* or Sierra Leone* or Somalia* or South Africa* or Sudan* or Swaziland * or Tanzania* or Togo * or Uganda* or Western Sahara Zambia * or Zimbabwe*).mp.

## Data extraction

Two independent reviewers (MW and MD) reviewed the titles, abstracts and keywords of every article retrieved by the search strategy according to the selection criteria that was developed. The full texts of the articles were retrieved for further assessment if the information given suggested that the study met the selection criteria or if there was any doubt regarding eligibility of the article based on the information given in the title and abstract. Endnote version X8 was used to manage search results and referencing.

## Assessment of methodological quality

Methodological quality of the included studies was assessed by two independent reviewers using a Critical Appraisal Skills Programme (CASP) checklist for quantitative studies [16]. The following criteria were used to assess the quality of the studies: clearly focused research question/objectives, appropriate method, clearly specified target population, adequate sampling techniques and sample size, adequate response rate, valid measurement tool, minimum selection bias, appropriate significance level and confidence interval, and relevance of the findings and applicability to our study. Finally, the quality of each paper was rated from an overall 10-point scale that was scored from 0 (none of the quality measures were met) to 10 (all quality measures were met). The quality of the paper was based on the sum of points awarded. Studies were rated as poor quality (score ≤6), medium quality (7–8), and high quality (≥9) (see S1 Table). Any disagreement between the reviewers was resolved by discussion to reach a consensus.

## Data analysis and synthesis

The extracted information was presented in summary form using tables and in narration. The eligibility criteria for studies to be included in the meta-analysis were studies that calculated odds ratios. In addition, in order to run meta-analysis for specific risk factors in this review, we set four and above studies per risk factor as a cut-off point. For risk factors that met the meta-analysis eligibility criteria, a random effects model was utilised to pool the effect sizes of the individual risk factors, taking into account between-study heterogeneity. Because of the low number of cohort and experimental studies (N = 8) for risk factors, we did not include the Relative Risk (RR) in the meta-analysis. The $I^2$ statistic was used to explain the between-study heterogeneity (0–100%), with higher percentage variation suggesting more heterogeneity or differences among studies. A test of the heterogeneity of each study data set was obtained for the different articles and showed the level of inconsistency ($I^2 > 50\%$), thereby warranting the use of a random effect model in all the meta-analyses. Forest plots were used to present the combined estimate with a 95% CI. Publication bias was assessed by an asymmetry test. The PRISMA checklist was employed to present the findings of studies on risk factors for preeclampsia and eclampsia among pregnant women in SSA [15]. All analyses were conducted in STATA version 16 software [17].

## Results

After searching the databases, 6813 articles were identified. After removing duplicate articles, 4805 articles were eligible for title and abstract review. From these articles, 120 articles were eligible for full text review. Finally, 51 articles were included in this systematic review and meta-analysis study (S1 Fig in S1 File).

## Description of included studies

Fifty-one studies met the inclusion criteria for systematic review (Table 1). Eleven countries were included in the studies, with the majority from Nigeria, Ethiopia, South Africa, and Sudan. More than 75% of the studies were published in the past 10 years. Of the 51 studies, 34 studies used a case-control design, ten were cross-sectional, six were cohort studies, and one study was experimental. The cohort studies and experimental study were not used for the meta-analysis of this paper due to the number of such studies being limited.

Overall, this review included 25,789 pregnant women in SSA countries. Of the 51 studies included in this study, 37 assessed preeclampsia, while four studies examined eclampsia and ten studies included both preeclampsia and eclampsia. Of these studies, based on the methodological quality assessment, only two studies [18, 19] were found to be of poor quality. We did not included those studies in meta- analysis. We included all 51 studies in the systematic review and out of those 16 studies in the meta-analysis review.

## Risk factors for preeclampsia/eclampsia

**I. Maternal age.**   Fourteen studies found that maternal age was a risk factor for preeclampsia and eclampsia [13, 21, 24, 25, 30, 32, 34, 46, 48, 49, 51, 53, 57, 64]. In this review, we identified four studies that had found a statistically significant association between older age and a higher risk of preeclampsia and eclampsia [25, 46, 48, 49, 57]. In contrast, three studies found younger mothers were more likely to develop preeclampsia and eclampsia as compared to older mothers [24, 30, 34]. One study conducted in Sudan showed both younger age and older age were associated with higher odds of preeclampsia and eclampsia [53].

A cross-sectional study conducted in Tanzania showed that mothers aged $\geq$ 35 years have 2.6 times more risk to develop preeclampsia than younger mothers (AOR: 2.6; 95% CI: 1.8, 3.7) [21]. Another study conducted in Ethiopia also found pregnant women aged $\geq$ 35 years were twice as likely to develop preeclampsia and eclampsia compared to younger mothers (AOR: 2.07; 95% CI:1.02, 4.19) [46]. Similar findings were observed in other studies conducted in Ethiopia and Sudan [48, 49].

On the other hand, studies conducted in Mozambique and Sudan found that mothers aged less than 18 and 20 years old were at greater risk of developing preeclampsia/eclampsia with the odds of 3.61 (OR: 3.61; 95% CI: 2.29, 5.68) and 7.6 (OR: 7.6; 95% CI: 2.9, 19.9) as compared to mothers aged greater than 35 years old, respectively [13, 53]. In contrast, women age less than 24 were found less risk for preeclampsia as compared to women age $\geq$ 35 years (AOR: 0.009; 95% CI (0.00, 0.31) [57]. A retrospective cohort study conducted in Ethiopia to assess the association between pre-partum and post-partum eclampsia showed that younger mothers were at significantly greater risk for pre-partum eclampsia as compared to older mothers [24, 30]. A five-year review of women with eclampsia conducted in Nigeria showed that teenagers (aged less than 19 years old) were at high risk for eclampsia [30].

Unlike the above studies, one study conducted in Sudan reported that both mothers in extreme of ages (age $\leq$ 20 and $\geq$ 35 years) were significantly at risk for preeclampsia with the odds of 7.6 (OR: 7.6; 95% CI: 2.9, 19.9) and 10.2 (OR: 10.2; 95% CI: 3.2, 32.2) as compared to their counterpart, respectively [53].

Note that, due to the different use of age categorisation among different studies, we did not run a meta-analysis for this variable. The age categorisation among the different studies varied. Most studies showed categorized younger age as being less than 20 years and older age was 35 years and above. Based on this review early age and later age of reproductive women were more at risk as compared to the other group age of women.

**Table 1. Summary table of studies included in risk factors for preeclampsia and eclampsia in SSA countries.**

| First author/ year | Country | Study design | Study size | Risk factor assessed |
|---|---|---|---|---|
| Allen Meeme et al. (2016) [20] | South Africa | Case-control study | 133 | Significantly lower Placental Growth Factor (PIGF) and higher placental anti-angiogenic factors soluble fms-like tyrosine kinase-1(SFLT1) found black African women diagnosed with preeclampsia as compared to normotensive pregnant women. In addition, in women with early onset preeclampsia, diastolic blood pressure and the level of (sFLT1) found significantly higher as compared to women with late onset preeclampsia. Furthermore, sFLT1/ PlGF ratio was significantly higher in women with preeclampsia than in normotensive pregnant women (p<0.05). |
| Dorah Mrema et al. (2018) [21] | Tanzania | Cross sectional study | 582 | The following factors were significantly associated with preeclampsia: overweight (Adjusted Odds Ratio (AOR): 1.4; 95% CI: 1.2, 1.8), obesity (AOR: 1.8; 95% CI (1.3, 2.4), maternal age (35–50 years) (AOR: 2.6; 95% CI: 1.8, 3.7) and mother's tribe (Pare) (AOR: 1.9; 95% CI: 1.4, 2.4). However, illiteracy (AOR: 1.2; 95% CI: 0.5, 3.1), second pregnancy (AOR: 0.7; 95% CI: 0.6, 0.9), mother living without partner (AOR: 0.8; 95% CI: 0.6, 1.3), mother's height < 155 cm (AOR: 0.7; 95% CI: 0.6, 1.0) and mother's height between 155–164 cm (AOR: 0.7 95% CI: 0.6, 0.9) were reported as a risk factors but not statistically significant. |
| Jocelynn T. Owusu et al. (2013) [22] | Ghana | Cross-sectional study | 220 | Women snoring during pregnancy was independently associated with preeclampsia (Odds Ratio (OR): 3.5; 95% CI: 1.4, 8.5). |
| Helen C. Okoye et al. (2016) [23] | Nigeria | Cross- sectional study | 180 | The mean free protein S antigen level was higher among women with preeclampsia than in the control group (P<0.004). No association was found between deficiencies of protein S antigens and preeclampsia. |
| Yifru B. and Gezahegn E. (2015) [24] | Ethiopia | Cohort study | 361 | Prepartum eclampsia were relatively younger age women (15–24), higher proteinuria, severe thrombocytopenia, started in preterm period, delivery takes place at health facility (82%) (P<0.0001)<br>Postpartum eclampsia high proportion of anemia, doubled SGOT level, higher home delivery without skilled assistance (81.6%) (P<0.0001). Pre and postpartum eclampsia were higher in both primiparous and multiparous. |
| Charlotte T. Nguefack et al. (2018) [25] | Cameroon | Cross- sectional study | 170 | Risk factors for Early onset preeclampsia: primiparous women (OR: 4.18; 95% CI: 1.82, 9.61), parity >4 (OR: 2.99, 95% CI: 0.94, 9.55), having a history of chronic hypertension (OR: 6.35; 95% CI: 1.24, 32.52) and history of preeclampsia (OR: 3.99; 95% CI: 1.62, 9.82), and having a family history of hypertension (OR: 1.37; 95% CI: 0.72, 2.59).<br>Risk factor for Late Onset preeclampsia: less income (OR: 3.05; 95% CI: 1.06, 8.80), not having the same father (OR: 2.10; 95% CI: 1.10, 4.02), having a family history of diabetes (OR: 0.71; 95% CI: 0.35, 1.43), having a family history of preeclampsia, having a sister with a preeclampsia/eclampsia history (OR: 1.96; 95% CI: 0.47, 8.16), and having a mother with a preeclampsia/eclampsia history (OR: 0.49; 95% CI: 0.05, 4.50).<br>No significant difference observed between the two groups with the following risk factors: Having kidney disease, drinking alcohol during pregnancy, maternal age, marital status, employment status, education status. |
| Teklit Grum et al. (2017) [14] | Ethiopia | Case-control study | 291 | Primigravida (AOR: 2.6; 95% CI: 1.38, 5.22), history of preeclampsia during prior pregnancy (AOR: 4.28; 95% CI: 1.61, 11.43), multiple pregnancy (twin) (AOR: 8.22; 95% CI: 2.97, 22.78), receiving nutritional counselling during pregnancy (AOR: 0.22; 95% CI: 0.1, 0.48) and drinking alcohol during pregnancy (AOR: 3.97; 95% CI: 1.8, 8.75). However, the following risk factor not statistically significant in this study: having a family history of hypertension (AOR: 2.24; 95% CI: 0.67, 7.46), occupation, being a student (AOR: 2.23; 95% CI: 0.67, 7.46) and a housewife (AOR: 1.72: 95% CI: 0.83, 3.58). In addition, unplanned pregnancy (AOR: 2.1; 95% CI: 0.68, 7.46), using hormonal contraceptive (AOR: 1.07; 95% CI: 0.52, 2.2), and non-fruit intake (AOR: 1.57; 95% CI: 0.47, 5.26). |
| M. A. Ikpen et al. (2012) [26] | Nigeria | Case-control study | 160 | Preeclampsia was associated with significant reductions in levels of vitamin C and E (p < 0.05). |

(*Continued*)

**Table 1.** (*Continued*)

| First author/ year | Country | Study design | Study size | Risk factor assessed |
|---|---|---|---|---|
| Mulualem Endeshaw et al. (2015) [27] | Ethiopia | Case-control Study | 453 | Daily coffee drinking during pregnancy (AOR: 1.78; 95% CI: 1.20, 3.05) and first trimester anaemia (AOR: 2.47, 95% CI: 1.12, 7.61) were risk factors. Whereas, consumption of fruit (AOR: 0.51; 95% CI: 0.29, 0.91) or vegetables (AOR: 0.46; 95% CI: 0.24, 0.90) least three times a week during pregnancy and compliance with folate intake during pregnancy (AOR: 0.16; 95% CI: 0.08, 0.29) found protective factors. In addition, for every 1 cm increase of MUAC, there was an increase in the incidence rate of pre-eclampsia (AOR: 1.35; 95% CI: 1.21, 1.51). |
| Teklit Grum et al. (2018) [28] | Ethiopia | Case-control study | 243 | Protective for preeclampsia or eclampsia were: Fruit intake during pregnancy (AOR: 0.94; 95% CI: 0.20, 4.32), vegetable intake during pregnancy (AOR: 0.95; 95% CI: 0.01, 0.71); and receiving nutritional counselling during antenatal care (AOR: 0.17; 95% CI: 0.05, 0.6). On the other hand, being a nulliparous (AOR: 2.02; 95% CI: 1.15, 3.55) and pregnancy interval < 1 year (AOR: 4.43; 95% CI: 1.78, 11.04) was a risk factor for preeclampsia or eclampsia. In addition having previous preeclampsia (AOR: 2.64; 95% CI: 0.77, 9.0). Coffee intake during pregnancy; daily (AOR: 3.26; 95% CI: 0.42, 25.36), at least once per week (AOR: 5.09; 95% CI: 0.93, 27.97) reported as risk but not statistically significant. |
| K. A. Frank et al. (2004) [29] | South Africa | Cohort study | 2,600 | No association between HIV seropositivity and the risk of developing preeclampsia/eclampsia reported in this study. |
| Ugochukwu V Okafor, et al. (2009) [18] | Nigeria | Cross-sectional study | 48 | Increased incidence of eclampsia during the rainy season than dry season. |
| SA Okogbenin, et al. (2010) [30] | Nigeria | Cross-sectional study | 78 | Eclampsia higher in teenagers and older women age >35 (P<0.0000) and higher incidence of eclampsia reported in primigravida and grand multiparity (>5 pregnancies). |
| Mahomed K., et al. (2007) [31] | Zimbabwe | Case-control study | 355 | Women in the highest quartile group for total omega-3 fatty acids experienced lower in risk of preeclampsia (OR: 0.86; 95% CI: 0.45, 1.63) as compared to women in the lowest quartile. A strong statistically significant positive association of diunsaturated fatty acids with a trans double bond with risk of preeclampsia was observed. Women in the upper quartile of 9-cis 12-trans octadecanoic acid (C18:2n6ct) three times higher risk of preeclampsia (OR: 3.02; 95% CI: 1.41, 6.45) compared with those in the lowest quartile experienced. |
| Gilles Guerrier, et al. (2013) [32] | Nigeria | Case-control study | 419 | Personal history of preeclampsia (AOR: 21.5; 95% CI: 14.2, 32.5), personal history of pre-existing hypertension (AOR: 10.5; 95% CI: 5.8, 19.0), primiparous (AOR: 2.5; 95% CI: 1.5, 4.3), housewife (AOR: 1.9; 95% CI: 1.2, 3.3), fewer than four antenatal care visits (AOR: 1.6; 95% CI: 1.1, 2.4), age less than 20 years old (AOR: 1.2; 95% CI: 0.7, 2.1) were significantly associated with severe preeclampsia/ eclampsia. Whereas, school attendance (AOR: 1.7; 95% CI: 0.9, 3.5), family history of preeclampsia (AOR: 1.0; 95% CI: 0.5, 2.1), family history of chronic hypertension (AOR: 1.2; 95% CI: 0.6, 2.3) and traditional treatment during pregnancy (AOR: 1.4; 95% CI: 0.9, 1.9) were identified as risk factor but not statistically significant. |
| Nadir A. Ahmed, et al. (2019) [33] | Sudan | Case-control study | 360 | The factor V Leiden-variation reported at positive associated with preeclampsia as compared to normotensive women (OR: 18.60; 95% CI: 2.38, 136.1). |
| Leonard O. Ajah, et al. (2016) [34] | Nigeria | Case-control study | 240 | Severe preeclampsia and eclampsia significantly common among adolescents, rural dwellers, poorly educated, unemployed, unbooked and nulliparous women (p-value <0.0001). |
| Grazyna A. Stanczuk, et al. (2007) [35] | Zimbabwe | Case-control study | 172 | Women who developed eclampsia/pre-eclampsia were more likely to have the high-producer allele T in codon 10 of the transforming growth factor-beta 1 gene as compared to normotensive women. |
| Kathleen M Powis, et al. (2013) [36] | Botswana | Experimental study | 560 | In HIV infected women with preeclampsia/eclampsia, placental growth factor (AOR 1.28; 95% 1.05, 1.66) and enrolment viral load of 100,000 copies/ml (AOR: 7.15; 95% CI: 1.35, 45.01) were associated with increased risk of preeclampsia/ eclampsia. |
| Dorothy J. Vanderjagt, et al. (2004) [37] | Nigeria | Case-control study | 173 | The urinary microalbumin/creatinine, protein/creatinine and serum homocysteine concentration values were markedly increased in the women with preeclampsia/eclampsia (p<0.001). However, mean level of cardioprotective HDL-cholesterol lower in women with preeclampsia/eclampsia |

(*Continued*)

**Table 1.** (*Continued*)

| First author/ year | Country | Study design | Study size | Risk factor assessed |
|---|---|---|---|---|
| Candice B. Roberts, et al. (2004) [38] | South Africa | Case-control study | 807 | No significant differences in the distribution of any of polymorphic variants were found between women with preeclampsia/eclampsia and normotensive women. |
| Jonah Musa, et al. (2018) [39] | Nigeria | Cohort study | 307 | Higher risk of preeclampsia were reported in women with previous history of preeclampsia (RR: 5.1; 95% CI: 2.2, 12.1) and BMI at booking ≥ 25 kg/m2 (RR: 3.9; 95% CI: 1.5, 10.0) as compared to healthy pregnant women. No statistical significant finding observed in women with having a previous miscarriage (RR: 0.9; 95% CI: 0.4, 1.90), nullparous (RR: 0.9; 95% CI: 0.5, 1.9) and HIV positive women (RR: 1.5; 95% CI: 0.4, 5.9). |
| Oladapo Olayemi, et al. (2010) [40] | Nigeria | Cohort study | 1850 | Same paternity abortions reduced the risk of hypertension (OR: 0.48; 95% CI: 0.31, 0.73). Previous abortions did not reduce the odds of hypertension in pregnancy (OR: 1.25; 95% CI: 0.83, 1.88). Rural dwelling reduced the odds of developing hypertension in pregnancy (OR: 0.54; 95% CI: 0.42, 0.70). Among those who reported same paternity abortions to the index pregnancy, the incidence of hypertension was 30.12%, while among those without same paternity abortions was 34.10%; this difference was not statistically significant (P < 0.164). |
| VMS Kalumba, et al. (2013) [41] | South Africa | Case-control study | 492 | HIV positive women were at lower risk of developing preeclampsia as compared to women without preeclampsia (OR: 0.62; 95% CI: 0.47, 0.82). |
| Aleksandar Rajkovic, et al. (2000) [42] | Zimbabwe | Case-control study | 356 | Women in the lowest quartile plasma folate experienced increased risk of preeclampsia as compared with women in the highest quartile (OR: 6.7; 95% CI: 3.0, 15.1). Each 1 nmol/L increase in folate corresponded to a 10% reduction in risk of preeclampsia (AOR: 0.9; 95% CI: 0.8, 0.9). No significant association found plasma vitamin B12 concentrations and risk preeclampsia. Women with low folate and the C/T genotype experienced doubled the risk of preeclampsia as compared to women with normal plasma folate and the C/C genotype, (AOR: 2.1; 95% CI: 0.8, 5.5). Positive association observed in low plasma folate concentrations and the C/C genotype and risk of preeclampsia (AOR: 3.9; 95% CI: 2.1, 7.2). |
| Rosemary J, et al. (2004) [43] | South Africa | Case-control study | 687 | More women with preeclampsia were homozygous for the T allele compared with control group (OR: 0.59; 95% CI: 0.42, 0.83). Heterozygotes occurred significantly higher rate among women with preeclampsia as compared with normotensive women (OR: 1.49; 95% CI: 1.07, 2.08). Allele frequencies also differed significantly between women preeclampsia and without preeclampsia (OR: 0.69; 95% CI: 0.53, 0.88). No differences in the 677T->C or 1298A->C MTHFR alleles were found between the study groups and controls; very few women were homozygous for either variant allele. |
| Babatunde Salako, et al. (2004) [44] | Nigeria | Cohort study | 100 | Statistically significant an increased i albumin excretion incidence was reported among women with preeclampsia (P value < 0.05). |
| Abdelmageed Elmugabil, et al. (2016) [45] | Sudan | Case-control study | 280 | Women who had absence antenatal care visits (OR: 2.75, 95% CI: 1.17, 6.49) and O blood group (OR: 1.78; 95% CI: 1.08, 2.93) were at higher risk of preeclampsia as compared to the controls. |
| Mulualem Endeshaw, et al. (2016) [46] | Ethiopia | Case-control study | 453 | For overall women with preeclampsia: MUAC ≥ 25 cm (AOR: 3.33; 95% CI 1.85, 5.79), fruit intake (AOR: 0.47; 95% CI 0.27, 0.83), drinking coffee during pregnancy (AOR: 2.14; 95% CI 1.32, 3.46), folate intake (AOR: 0.18; 95% CI 0.10, 0.32) were factors. When stratified with early and late onset preeclampsia, folate intake (AOR: 0.12; 95% CI 0.04, 0.33) found protective from preeclampsia. Whereas, MUAC ≥ 25 cm (AOR: 3.63; 95% CI 1.89, 6.97), fruit intake (AOR: 0.34; 95% CI 0.16, 0.71) and folate intake (AOR: 0.16; 95% CI 0.08, 0.33) were factors associated with late onset preeclampsia. |

(*Continued*)

**Table 1.** (Continued)

| First author/ year | Country | Study design | Study size | Risk factor assessed |
|---|---|---|---|---|
| Oluranti B. Familoni, et al. (2004) [47] | Nigeria | Retrospective cross-sectional study | 127 | The group with preeclampsia/eclampsia tended to be younger (25.8 ± 5.8 vs. 29.9 + 4.6, p<0.092) and appeared to have higher systolic BPs (176.06 ± 24.10 vs. 160.19 ± 27.04, p<0.089), although was not statistically significant. There was no significant difference in the diastolic pressures of the three groups. The preeclampsia/eclampsia group also belonged more to the higher-risk group (63.6 vs. 31.3, p<0.0006), while the patients with gestational HTN belonged more to the low-risk group (68.8 vs. 36.4, p<0.0006). When logistic regression was used, although statistically not significant, the trend observed was that the older the patient (OR: 0.36; 95% CI: 0.12, 1.07) and the more literate the patient (OR: 0.42; 95% CI: 0.14, 1.18) the less likely they were to be in the high-risk group. The parity of the patient also had no effect on whether she belonged to the high- or low-risk group (OR: 0.79, 95% CI: 0.26, 2.39). Similarly, regular ANC attendance appeared to reduce the likelihood that a patient belongs to the high-risk group (OR: 0.58; 95% CI: 0.16, 2.09). There appeared to be at least twice the risk of developing eclampsia (OR: 2.37; 95% CI: 0.85, 6.66) and abruptio (OR: 2.00, 95% CI: 0.17, 23.29) in patients who belonged to the high-risk group. |
| Ishag Adam, et al. (2013) [48] | Sudan | Retrospective case-control study | 1765 | The risk factors for preeclampsia were: women aged >35 years (OR: 1.4; 95% CI: 1.1, 1.8), primiparity (OR: 3.3; 95% CI: 2.7, 4.0), multiparous (>5birth) (OR: 3.1; 95% CI: 2.4, 4.0), and anaemia during pregnancy (OR: 3.3; 95% CI: 2.8, 3.9). In addition, increased risk of preeclampsia were observed in illiterate women (AOR: 2.6, 95% CI: 2.0, 3.5) and absence of ANC visits (AOR: 4.2; 95% CI: 2.9, 6.0). Placenta previa found significant protective factor of preeclampsia (OR:0.3; 95% CI: 0.1, 0.7). |
| Gizachew Assefa Tessema, et al. (2015) [49] | Ethiopia | Hospital-based cross-sectional study | 490 | Women having a family history of hypertension (AOR: 7.19; 95% CI: 3.24, 15.2), chronic hypertension (AOR: 4.3; 95% CI: 1.33, 13.9), age ≥35 years (AOR: 4.5; 95% CI: 1.56, 12.8), family history of diabetes mellitus (AOR: 2.4; 95% CI: 1.09, 5.6) and being unmarried (AOR: 3.03; 95% CI: 1.12, 8.2) were found to be associated with preeclampsia. |
| Jurgen Wacker, et al. (2000) [50] | Zimbabwe | Prospective observational study (cohort) | 210 | Women with riboflavin deficiency were more likely to develop preeclampsia as compared to the riboflavin-adequate women (OR: 4.7; 95% CI: 1.8, 12.2). Significant lower intracellular free flavin adenine dinucleotide reported in women with preeclampsia compared to normal pregnant women. |
| Rose I. Anorlu, et al. (2005) [51] | Nigeria | Case-control study | 408 | The risk factors that were associated with increased risk of preeclampsia were: nulliparity (AOR: 4.77; 95% CI: 2.90, 7.78), stressful work during pregnancy (AOR: 2.10; 95% CI: 1.20, 3.71), stressful home environment (AOR: 1.97; 95% CI:1.27, 3.69), previous pre-eclampsia (AOR: 11.68; 95% CI: 3.81, 37.61), history of chronic hypertension (AOR: 2.21; 95% CI: 1.17, 6.20), a body weight greater than 80 kg (AOR: 2.01; 95% CI: 1.05, 3.87), having a family history of hypertension (AOR: 2.21; 95% CI: 1.17, 6.20) and multiple pregnancy (AOR: 2.71; 95% CI: 1.27, 6.13). Maternal age < 19 yeas (AOR: 3.01; 95% CI: (0.51, 20.99), educated mother (AOR: 1.55; 95% CI: 0.67, 3.71), senior occupation (AOR: 1.93; 95% CI: 0.86, 4.52), changing partner (AOR: 1.54; 95% CI: 0.21, 9.67), and having chronic hypertension (AOR: 2.72; 95% CI: 0.72, 2.09) were associated with preeclampsia but not statistically significant. |
| Paul Kiondo, et al. (2012) [52] | Uganda | Case-control study | 559 | Significantly high risk of preeclampsia were reported in the following factors: low plasma vitamin C (AOR: 3.19; 95% CI: 1.54, 6.61), low education level (AOR: 1.67; 95% CI: 1.12, 2.48), chronic hypertension (AOR: 2.29; 95% CI: 1.12, 4.66), family history of hypertension (AOR: 2.25; 95% CI: 1.53, 3.31), primiparity (AOR: 2.76; 95% CI: 1.84, 4.15), parity > 5 (AOR: 3.71; 95% CI: 1.84, 7.45) and drinking alcohol during pregnancy (AOR: 1.65; 95% CI: 0.93, 2.94). |
| António Bugalho, et al. (2001) [13] | Mozambique | Case- control study | 526 | Risk factors for eclampsia were age < 18 years (OR: 3.61; 95% CI: 2.29, 5.68), household size < 3 individuals (OR: 2.16; 95% CI: 1.30, 3.59), unwanted pregnancy (OR: 1.75; 95% CI: 1.15, 2.66), walking to ANC clinics (OR: 4.31; 95% CI: 2.77, 6.72), being a housewife (OR: 1.27; 95% CI: 0.71, 2.28), being illiterate (OR: 0.66; 95% CI: 0.40, 1.09), unplanned pregnancy (OR: 0.75; 95% CI: 1.15, 2.66), and having chronic hypertension (OR: 2.04; 95% CI: 1.31, 3.17). |

*(Continued)*

**Table 1.** (*Continued*)

| First author/ year | Country | Study design | Study size | Risk factor assessed |
|---|---|---|---|---|
| Annelies Immink, et al. (2008) [19] | South Africa | Retrospective cross-sectional study | 1329 | Women admitted to the hospital in winter had a higher risk of developing preeclampsia compared to those admitted in summer (OR: 1.69; 95% CI: 1.07, 1.53). The risk of developing preeclampsia in June was nearly three times higher than in February (summer in South Africa, reference month) (OR: 2.81; 95% CI: 2.06, 3.83). There was a significant correlation between the number of admissions with preeclampsia and the minimum daily temperature ($p<0.032$). |
| AbdelAziem A Ali, et al. (2011) [53] | Sudan | Retrospective case-control study | 4315 | The corrected risk for preeclampsia increased only in severe anaemia (AOR: 3.6; 95% CI: 1.4, 9.1). The prevalence of preeclampsia and eclampsia was significantly higher in women with severe anaemia (8.2% and 3.3%, respectively). The corrected risk for preeclampsia (OR: 3.6; 95% CI: 1.4, 9.1) increased only in severe anaemia. Being younger ($<$ 20 years) and older ($>$ 35 years) women were associated with increased risk for preeclampsia/eclampsia (OR: 7.6; 95% CI: 2.9, 19.9) and (OR: 10.2; 95% CI: 3.2, 32.2), respectively. |
| Mohamed A. Ahmed, et al. (2019) [54] | Sudan | Case-control | 186 | H. pylori sero-positivity and absence of ANC visits were risk factors for preeclampsia (AOR: 4.933 95% CI: 2.08, 11.69) and (OR: 14.17; 95% CI: 5.28, 38.02), respectively. |
| Sumia F. Ahmed, et al (2020) [55] | Sudan | Case-control | 320 | The proportion of the T allele was significantly higher in women with preeclampsia than in healthy pregnant women (OR: 9.3: 95% CI: 2.7, 31.2). MTHFR C677T gene polymorphism found associated with the Increased risk of preeclampsia (OR: 10.1 95% CI: 3.0: 34.2). |
| Margaret O. Alese, et al. (2019) [56] | South Africa | Case-control | | Irrespective of HIV status, there was a significantly lower protein concentration of the analytes in women with preeclmpsia as compared to normotensive women. In addition, according to HIV status there was no significant difference in expression of ERK1/2, p38MAPK and p90 RSK prosurvival markers between women with preeclampsia and without preeclampsia. |
| Alemayehu Sayih Belay and Tofik Wudad (2019) [57] | Ethiopia | Cross-sectional | 129 | Women age $\leq$ 24 years (AOR: 0.009; 95% CI: 0.001,0 .31), single birth pregnancy (AOR0 .07; 95% CI: 0.007, 0.77), no history of DM (AOR: 0 .06; 95% CI: 0.007, 0.46) were protective factor for preeclampsia. Whereas, rural dweller (AOR: 5.04; 95% CI:0.670–37.96), had more than one partner (AOR: 1.57; 95% CI:0.15, 16.51), had family history of DM (AOR:1.63; 95% CI0.09, 29.35), history of kidney disease (AOR: 1.34; 95% CI: 0.18, 9.89), had ANC in previous pregnancy (AOR:0.26; 95% CI:0.02, 3.15) and parity $<$ 2 (AOR: 2.55; 95% CI 0.24, 26.56) were reported as risk factor but not statistically significant. |
| Husham O. Elzein et al. (2019) [58] | Sudan | Case-control | 100 | Women with severe PE were found to have a significant difference in Factor V Leiden mutated gene (OR: 20.20; 95% CI: 1.132–360.5) and FII G20210A mutation (OR: 17.41; 95% CI: 0.96, 314.0) as compared to normotensive women. |
| Hameed M. Hamid et al. (2020) [59] | Sudan | Case-control | 120 | For rs3025039, CT, CT+TT, and the T allele were risk factors for preeclampsia. Regarding rs16944, only the heterozygous genotype CT was associated with preeclampsia. Regarding rs1143634, CT, CT+TT, and the T allele were risk factors for preeclampsia |
| Selma Mahmoud, et al. (2019) [16] | Sudan | Case-control | 90 | Significant higher level of migration inhibitory factor (MIF) and lower level Insulin like growth factors (IGF-1) in the women with preeclampsia as compared to normotensive women. |
| Collins E. M. Okoror, et al. (2020) [60] | Nigeria | Case-control | 81 | Women with hypocalcaemia more likely to develop preeclampsia (AOR: 7.63; 95% CI 1.64, 35.37) compared to those with normal serum calcium. Pregnant women of social classes 2 and 3 compared to social class 5 were less likely to develop pre- eclampsia (AOR: 0.01, 95% C: 0.001, 0.46 and (AOR: 0.01; 95% CI 0.001, 0.24). |
| F. D. H. Olalere, et al. (2018) [61] | Nigeria | Case-control | 240 | Mean levels of triglycerides; low-density lipoprotein; high-density lipoprotein and very low-density lipoprotein were higher than normal in both with and without preeclampsia. Maternal serum total cholesterol TC, TG, and LDL was significantly higher in severe, compared to mild preeclampsia ($p < .001$). |
| V. O. Osunkalu, et al. (2020) [62] | Nigeria | Case-control | 400 | Occurrence of preeclampsia was significantly associated with presence of T allele of MTHFR (OR: 1.85; $p < 0.05$) and G allele of MTR genes (OR: 1.27; $p < 0.05$), Homozygosity of TG haplotype significantly increased the occurrence of preeclampsia (OR = 2.252; $p < 0.05$) |

(*Continued*)

**Table 1.** (Continued)

| First author/ year | Country | Study design | Study size | Risk factor assessed |
|---|---|---|---|---|
| Semone Thakoordeen-Reddy, et al. (2020) [63] | South Africa | Case-control | 428 | A significant association between the maternal APOL1 G1 risk allele and early onset preeclampsia reported (OR 2.2, p < 0.03). Among women with early onset preeclampsia, 5% (OR: 0.94; 95%CI: 0.29, 3.12) of the study population carried two risk alleles, 49% OR: 1.34; 95% CI: 0.77, 2.3) carried at least one risk allele, while 46% of the participants did not carry either risk allele, compared to the normotensive pregnant group. |
| J. Wandabwa, et al. (2010) [52] | Uganda | Case-control | 643 | A risk factor identifies for preeclampsia were: distance from home to health facility >10 km (AOR: 3.8; 95% CI; 1.9, 7.7), low socio—economic status (AOR: 7.6; 95% CI: 3.9, 26.9), chronic hypertension (AOR: 26.9; 95% CI: 4.3, 170.4), family history of hypertension (AOR: 1.9; 95% CI: 1.2, 2.9), null-parity (AOR: 2.2; 95% CI: 1.2, 4.3), delivery of male babies (AOR: 1.5; 95% CI: 1.0, 2.3), previous abortion (AOR: 2.6; 95% CI: 1.3, 5.1), having previous preeclampsia (AOR: 2.6; 95% CI:1.0, 6.6), not blood pressure check during ANC visits (AOR: 2.5; 95% CI:1.2, 5.2), birth spacing >60 months (AOR: 8.3; 95% CI:2.6, 26.4) and nascence of ANC visits (AOR:3.4; 955 CI: 1.4, 8.5) |

**II. Previous preeclampsia/eclampsia.** Six studies were identified that investigated the relationship between having previous preeclampsia or eclampsia with the risk of preeclampsia or eclampsia in the present pregnancy [14, 25, 28, 32, 51, 52]. Five of the six studies reported that having a history of preeclampsia or eclampsia in a previous pregnancy increased the tendency of recurrence in the present pregnancy [14, 25, 32, 51, 52]. A study conducted in Ethiopia found that a mother with a history of preeclampsia in her previous pregnancies was four times more likely to develop preeclampsia and eclampsia in subsequent pregnancies (AOR: 4.28; 95% CI: 1.61, 11.43) [14]. Similarly, a study conducted in Cameroon reported that having preeclampsia/eclampsia previously increased the likelihood of having early onset preeclampsia by four times as compared to late onset preeclampsia (OR: 3.99; 95% CI: 1.62, 9.82) [25]. Similar findings were reported in studies conducted in Nigeria and Uganda [32, 51, 52].

For this variable, all six studies met the inclusion criteria for meta-analysis. The meta-analysis finding revealed a strong significant association between having preeclampsia/eclampsia in previous pregnancies and development of preeclampsia or eclampsia in the current pregnancy. Based on pooled estimate of OR, women who had a history of preeclampsia or eclampsia were nearly six times higher risk of developing preeclampsia or eclampsia as compared to women who had no history of preeclampsia or eclampsia in past pregnancies (OR: 5.6; 95% CI: 1.82, 9.38). Heterogeneity was checked and a random effect model was used in this analysis (S2 Fig in S1 File). Both the Funnel plot and the Egger's test showed that there was no publication bias in the included studies.

**III. Family history of preeclampsia/eclampsia.** Eight studies were found that examined the association of having a family history of preeclampsia or eclampsia and the risk of developing preeclampsia or eclampsia [14, 25, 32, 39, 49, 52, 64, 65]. Five studies found that having a family history of preeclampsia or eclampsia was positively associated with the chance of developing preeclampsia or eclampsia in pregnant women [39, 49, 52, 64, 65]. For instance, a study conducted in Uganda showed that pregnant women who have a family history of hypertension were more than two times more likely to develop preeclampsia/eclampsia when compared to pregnant women who do not have a family history of hypertension (AOR: 2.25; 95% CI:1.53, 3.31) [52]. In addition, a cross-sectional study in Ethiopia found that a family history of hypertension during pregnancy increased the probability of recurrence in pregnant women by seven times (AOR: 7.19; 95% CI (3.24, 15.2) [49].

Seven out of eight studies were included in the meta-analysis of family history of pre-eclampsia or eclampsia as a risk factor for subsequent preeclampsia or eclampsia [14, 25, 32, 49, 52, 64]. Heterogeneity was checked and fixed effect model was used in this analysis. The pooled odds ratio showed that women with a family history of preeclampsia or eclampsia were 1.68 times more likely to develop preeclampsia or eclampsia during pregnancy (OR: 1.68; 95% CI: 1.26, 2.11) (S3 Fig in S1 File).

**IV. Maternal body mass index.** Five studies analysed the risk of preeclampsia or eclampsia in women with a high Body Mass Index (BMI) and all of them reported that pregnant women with a high BMI were at greater risk of both preeclampsia and eclampsia [21, 27, 39, 46, 51]. A study conducted among the 17,738 singleton births in Tanzania found that women who were overweight and women who were obese (BMI 25–29.9 and >30 kg/m$^2$, respectively) were 1.4 (OR: 1.4; 95% CI: 1.2, 1.8) and 1.8 (OR: 1.8; 95% CI: 1.3, 2.4) times more likely to have preeclampsia compared with women with a normal BMI, respectively [21]. The same study found that underweight women were 30% less likely to develop preeclampsia (AOR: 0.7; 95% CI: 0.4, 1.1) as compared to women with a normal BMI [21]. A case-control study conducted in Ethiopia showed that for every 1 cm increase of Mid Upper Arm Circumference (MUAC), there was an increase in the prevalence rate of preeclampsia by a factor of 1.4 (AOR: 1.35; 95% CI: 1.21, 1.5) [27]. Similarly, a result from a cohort study of 307 pregnant women in Nigeria found that a BMI > 25 kg/m$^2$ estimated at ≤ 20 weeks of gestation reported a four times higher risk and hazard of developing preeclampsia during the gestational follow-up time (RR: 3.9; 95% CI: 1.5, 10.0) [39].

Four studies were included in the meta-analysis and the pooled odds ratio analysis showed that women's BMI was associated with preeclampsia and eclampsia. The results of the Egger's test and Funnel plot showed there was no publication bias among the included studies. A random effect model was used for this variable to test the heterogeneity. The finding of the meta-analysis showed that women that have a higher BMI were nearly twice as likely to develop preeclampsia/eclampsia as women with a healthy BMI (OR: 1.69; 95% CI: 1.17, 2.21) (S4 Fig in S1 File).

**V. Chronic hypertension and diabetes mellitus.** In this review, having a pre-existing medical condition was an important predictor in the development of both preeclampsia and eclampsia. Studies conducted in different SSA countries showed the likelihood of developing preeclampsia and eclampsia was two to six times more likely among pregnant mothers with chronic hypertension compared to healthy pregnant mothers [13, 25, 32, 49, 51, 52]. In relation to this, Anorlu et al. (2005) found that the risk of preeclampsia increased more than two times among Nigerian pregnant women with a family history of hypertension (AOR: 2.21; 95% CI: 1.17, 6.20) [51]. In addition, pregnant women with no history of diabetes mellitus found a lower risk for developing as compared to women with diabetes mellitus history (AOR: 0.06; 95% CI: 0.007–0.47) [57]. Likewise, having diabetes mellitus in a family member was also found to significantly increase the risk of preeclampsia/eclampsia. A study conducted in Ethiopia showed that having a family history of diabetes mellitus doubled the odds of developing preeclampsia during pregnancy (AOR: 2.4; 95% CI: 1.09, 5.6) [49].

Overall, we included six studies in a meta-analysis that analysed pregnant women with chronic hypertension, as papers that examined other chronic health conditions did not meet the criteria to undertake meta-analysis. A fixed effect model was used to calculate the pooled odds ratio as $I^2$ <50%. Both Funnel plot and Egger's test revealed no publication bias. The pooled odds ratio showed mothers with chronic hypertension increased two fold the odds of having preeclampsia/eclampsia as compares to women without chronic hypertension (OR: 2.26; 95% CI: 1.49, 3.03) (S5 Fig in S1 File).

**VI. Anaemia during pregnancy.** Anaemia during pregnancy was positively associated with the probability of developing preeclampsia/eclampsia in five publications [24, 27, 46, 48, 53]. In this review, four of the studies mentioned the cut off point for presence of anaemia as hemoglobin level <11 g/dL [27, 46, 48, 53], whereas one study report hemoglobin level <12 g/dL as a diagnosis criteria for anaemia [24]. A case-control study conducted on 151 cases and 302 controls of pregnant women in Ethiopia found that women who had anaemia during the first trimester were 2.47 times more likely to develop preeclampsia compared to women without anaemia (AOR: 2.47; 95% CI: 1.12, 7.61) [27]. Another study in Ethiopia also reported that pregnant women with anaemia were nearly twice as likely to develop overall preeclampsia (AOR: 1.89; 95% CI: 0.64, 5.61) and three times more likely to develop late onset preeclampsia (AOR: 2.97; 95% CI: 0.91, 9.69) as compared to women without anaemia during pregnancy [46]. Similarly, a study conducted in Sudan revealed that pregnant women with anaemia had about a four-fold increased probability of developing preeclampsia as compared to women without anaemia (AOR: 3.6; 95% CI: 1.4, 9.1) [53]. Another study conducted in Ethiopia on clinical and biomarker differences in pre-partum and postpartum eclampsia found that a significantly high proportion of anaemia was observed among women with postpartum eclampsia (P<0.0001) [24]. The overall pooled effect showed that anaemia during pregnancy increased the risk of having preeclampsia/eclampsia more than three times as compared to mothers without anaemia (OR: 3.22; 95% CI: 2.70, 3.75) (S6 Fig in S1 File).

**VII. Maternal education.** This review found seven studies that analysed the relationship between preeclampsia and eclampsia and maternal education level. However, results of the association between maternal education and risk for hypertensive disorder of pregnancy were mixed. Four studies found that illiteracy/low-educational achievement were associated with higher odds of preeclampsia/eclampsia [34, 48, 52, 66]. Two of the studies found preeclampsia and eclampsia were statistically significantly associated with the risk of low maternal education [48, 52]. In contrast, three studies found that educated mothers had association with preeclampsia/eclampsia, but none of these studies found a significant result [25, 32, 51].

All seven studies were included in a meta-analysis and both the Funnel plot and the Egger's test showed that there was no publication bias in the included studies. The pooled estimated odds ratio showed that the level of maternal education was 1.12 (OR: 1.12; 95% CI: 0.59, 1.65). This indicates that the level of maternal education was not associated with preeclampsia and eclampsia as shown the confidence interval cross one (S7 Fig in S1 File).

**VIII. Nutrition and related factors.** In this review, counselling about nutrition and diet during ANC visits were found to be protective against preeclampsia and eclampsia. A study conducted in Ethiopia found women getting counselling about nutrition and dietary diversity during pregnancy reduced the risk of preeclampsia by 22% as compared to women with no nutrition counselling (AOR: 0.22; 95% CI: 0.1, 0.48) [14]. A similar finding was reported in a study conducted in Ethiopia (AOR: 0.17; 95% CI: 0.05, 0.6) [28]. When we examined food composition, different researchers reported eating fruit while pregnant reduced the risk of preeclampsia and eclampsia by almost up to half [14, 27, 28, 46]. For example, a study conducted in Ethiopia revealed that consumption of fruit at least three times a week during pregnancy was found to be protective against preeclampsia and reduced the risk by half as compared to inadequate fruit intake during pregnancy (AOR: 0.51; 95% CI: 0.29, 0.91) [27].

Vegetable consumption during pregnancy was also found to be a protective factor for preeclampsia/eclampsia [27, 28, 46]. For instance, a study conducted in Ethiopia found that women eating vegetables during her pregnancy reduced the risk of preeclampsia by almost half as compared to inadequate vegetable intake during pregnancy (AOR: 0.46, 95% CI: 0.24, 0.90) [27].

Other nutrition related factors were also reported to have a relationship with the risk of pre-eclampsia and eclampsia. A study conducted in Ethiopia by Mulualem et al. found that compliance to folate supplementation during pregnancy was associated with a reduced risk of preeclampsia of about 20% (AOR: 0.18; 95% CI: 0.10, 0.32) [46]. Whereas, Aleksandar et al. found that the lowest quartile plasma folate in pregnant women increased the risk of pre-eclampsia by 10 times as compared to the higher quartile plasma folate concentration (AOR: 10.4; 95% CI: 3.8, 28.3) [42]. The same study also reported, in each 1 nmol/L increase in folate corresponded to a 10% reduction in risk of preeclampsia (AOR: 0.9; 95% CI: 0.8, 0.9) [42]. Researchers found that pregnant women with riboflavin deficiency as compared to than in the riboflavin-adequate women and low plasma vitamin C as compared to adequate plasma vitamin C were more likely to develop preeclampsia, with an odds ratio of (OR: 4.7; 95% CI: 1.8, 12.2) and (AOR: 3.19; 95% CI: 1.54, 6.61), respectively [50, 52]. In addition, pregnant women in the highest quartile group for total omega-3 fatty acids compared with women in the lowest quartile experienced a 14% reduction in the risk of preeclampsia (OR: 0.86; 95% CI: 0.45, 1.63) [31]. Furthermore, women with hypocalcemia were about eight times more likely to develop preeclampsia compared to those with normal mean serum calcium ($< 8.6$ mg/dL) (AOR: 7.63; 95% CI: 1.64, 35.37) [60]. Another study showed that drinking coffee during pregnancy significantly increased the risk of preeclampsia/eclampsia by about two times compared to not drinking coffee during pregnancy (AOR: 1.78; 95% CI: 1.20, 3.05) [46]. We did not run meta-analyses for nutrition and related factors due to the variability among the individual factors.

**IX. Drinking alcohol during pregnancy.**   In this review, four studies investigated the relationship between maternal alcohol drinking during pregnancy and the risk of preeclampsia/eclampsia [14, 25, 46, 52]. Based on this review, none of the included studies reported the quantifying level of alcohol consumption in their reports. Only one study (conducted in Ethiopia) showed a significant association between alcohol use in pregnancy and eclampsia or pre-eclampsia; women who drank alcohol during pregnancy were four times more likely to develop preeclampsia or eclampsia than women who drank no alcohol while pregnant (AOR: 3.97; 95% CI: 1.8, 8.75) [14]. The overall pooled meta-analysis found that there was no significant difference between women drinking alcohol during pregnancy and the risk of preeclampsia and eclampsia (OR: 1.38; 95% CI: 0.70, 2.06) (S8 Fig in S1 File).

**X. Antenatal Care (ANC) visits.**   This review found eight studies that analysed the relationship between preeclampsia and eclampsia and having ANC visits [13, 32, 34, 45, 48, 52, 57, 64]. Five out of eight studies reported significant association between lack of ANC visits and preeclampsia/eclampsia [32, 34, 45, 48, 52]. Studies conducted in Nigeria, Sudan and Uganda showed that pregnant mothers with no ANC visits had higher odds of having preeclampsia/eclampsia as compared to pregnant women who did receive ANC services [32, 34, 45, 48, 52]. For instance, Ishang Adam et al. found that women with no ANC visits had four times increased odds of preeclampsia/eclampsia as compared to women with ANC visits (AOR: 4.2; 95% CI: 2.9, 6.0) [48]. Similarly, women without ANC visits in present pregnancy were 3.4 times higher as compared to those women having ANC visits (AOR: 3.4; 95% CI: 1.4, 8.5) [52]. Furthermore, a study conducted in Mozambique showed that women without transport access or had walk to reach health facility for ANC visits were four times more likely to develop eclampsia than women with transport access (OR: 4.31; 95% CI: 2.77, 6.72) [13].

For meta-analysis, we included six studies and both the Funnel plot and the Egger's test showed that there was no publication bias in the included studies. A random effect model was used to calculate the pooled odds ratio. The pooled odds ratio showed mothers had no ANC visits during pregnancy increased nearly threefold the odds of having preeclampsia/eclampsia as compares to women had ANC visits (OR:2.71; 95% CI: 1.45, 3.96) (S9 Fig in S1 File).

**XI. Parity.** The fourteen studies were found that investigated the association of number of pregnancy and the risk of developing preeclampsia or eclampsia [14, 21, 24, 25, 28, 34, 39, 48, 51, 52, 57, 64, 65]. Out of these, five studies analysed the risk of preeclampsia in primiparous women [14, 24, 25, 32, 64] and four of them reported increased risk of preeclampsia and eclampsia [14, 24, 25, 32]. Whereas, five studies reported the likelihood of preeclampsia and eclampsia in nulliparous [28, 34, 39, 51, 52] and four of them found significantly higher in nulliparous women as compared to multiparous women [28, 34, 51, 52]. In addition, two studies conducted in Sudan and Uganda reported both primiparous and multiparous women ($\geq 4$ pregnancies) were at increased risk of preeclampsia and eclampsia as compared to women with 2–4 pregnancies [48, 52]. In a study conducted in Tanzania, the risk of preeclampsia/ eclampsia found 30% less in the second pregnancy as compared to the first pregnancy (AOR: 0.7; 95% CI 0.6, 0.9) [21].

Overall, we included six studies in a meta-analysis that analysed primiparous women, as papers that examined other nulliparous and multiparous did not meet the criteria to undertake meta-analysis. A random effect model was used to calculate the pooled odds ratio. Both Funnel plot and Egger's test revealed no publication bias. The pooled odds ratio showed primiparous women 2.5 times more likely to develop preeclampsia/eclampsia as compared to multiparous women (OR: 2.52; 95% CI: 1.19, 3.86) (S10 Fig in S1 File).

**XI. Pregnancy related risk factors for preeclampsia/eclampsia.** This review found different pregnancy related risk factors, which were associated with the development of preeclampsia and eclampsia. Three studies reported a strong association between multiple births and the risk of developing preeclampsia and eclampsia [14, 51, 57]. For instance, a case-control study conducted in Nigeria showed that multiple pregnancy increased the odds of preeclampsia/eclampsia nearly threefold as compared to singleton pregnancy (AOR: 2.71; 95% CI: 1.27, 6.13) [51]. Similarly, a cross-sectional study conducted in Ethiopia reported that women with single pregnancy were protective against preeclampsia as compared to women with multiple birth (AOR: 0.071; 95% CI 0.007, 0.77) [57].

A study conducted in Ethiopia showed that less than a year's interval between pregnancies increased the chance of the mother developing preeclampsia/eclampsia by more than four times as compared to greater than a year's interval between pregnancies (AOR: 4.43; 95% CI: 1.78, 11.04) [28]. In contrast, a study conducted in Uganda showed that pregnancy interval more than 60 months increase the risk of preeclampsia as compared to less than 37 months (AOR: 8.3; 95% CI:2.6, 26.4) [52]. In addition, this study also revealed having previous abortion history increase the risk of preeclampsia and eclampsia [52]. Whereas another study reported, women had the same paternity induced abortion about 50% reduced the risk of hypertension in present pregnancy as compared to women without same paternity induced abortion (OR: 0.48; 95% CI: 0.31,0.73), and thus this finding support hypothesis the role of immunological factors in the pathophysiological mechanism preeclampsia [40]. Furthermore, a case-control study conducted in Mozambique showed that a woman who reported that her pregnancy was unwanted was 1.75 times more likely to develop preeclampsia/eclampsia as compared to women who reported that their pregnancy was wanted (OR: 1.75; 95% CI: 1.15, 2.66) [13]. For these risk factors, an insufficient number of studies had been conducted, so meta-analyses could not be conducted.

In this review, the following factors were identified as a risk factors for preeclampsia and eclampsia: paternity [25, 40], marital status [49], HIV/AIDS [36, 41], place of residence [34, 40, 57], family size [13, 51], maternal sleeping position [22], blood type [45, 64], H. pylori seropositivity [16], distance from health facility [52], placental malaria [64] and seasonal variation [19]. In addition, recently several researchers investigated the genetic components and related predisposing factors for preeclampsia and eclampsia in African women [16, 20, 23, 31, 33, 35,

37, 43, 55, 56, 59, 61–63]. However, this systematic review was not found sufficient evidence to draw strong conclusion to consider them as a risk factor for preeclampsia/eclampsia.

## Discussion

After meta-analysis and systematic review of the studies, being primiparous, a previous history of preeclampsia, a family history of preeclampsia/eclampsia, high BMI, chronic hypertension, anaemia during pregnancy and lack of ANC visits were the pertinent factors that were associated with preeclampsia/eclampsia. In this study, we aimed to synthesise the evidence on the risk factors of preeclampsia/eclampsia using published research since 2000 in SSA countries. We also aimed to determine the pooled odds ratios of different risk factors for preeclampsia/ eclampsia. In this review, we found the primiparous women at more than two time higher risk for preeclampsia/eclampsia. This result consistent with a systematic review conducted by Luo ZC et al and in their finding primiparous women had 2.4 time more likely to develop pre-eclampsia as compared to multiparous women (OR:2.42; 95% CI: 2.16, 2.71) [67]. There are also several studies reported consistent with our study finding [68–70]. More risk of pre-eclampsia/eclampsia in primiparous women can be explained by maternal immunological incompetency with the foetal tissue and this exposure might increase the risk of having pre-eclampsia/eclampsia in first pregnancy [10]. First time pregnant, mothers are a higher risk for developing serious complication including preeclampsia/ eclampsia. Therefore, screening and proper follow up of primiparous women during the course of first pregnancy to reduce the risk of preeclampsia and its complications.

A previous history of preeclampsia was strongly associated with risk of preeclampsia/ eclampsia in subsequent pregnancies. This was consistent with a systematic review of con-trolled studies where a seven-fold increase in the risk for women with previous preeclampsia compared with women with no previous preeclampsia was reported [71]. Another study con-ducted on primiparous mothers also showed that the risk of having preeclampsia/eclampsia could be up to 15% higher in their second pregnancy for women who had preeclampsia in their previous pregnancy, reaching 32% for women who had preeclampsia in their previous two pregnancies [67]. This strong association can be explained by the recurrent nature of the disease, in that it has a systemic effect on maternal organs. In order to reduce complications, women who previously experienced preeclampsia need special care when they become preg-nant again. Women with family histories of preeclampsia/eclampsia were noted to be at an increased risk for development of preeclampsia/eclampsia in the subsequent pregnancies. We found in a meta-analysis of five studies that family history of preeclampsia/eclampsia increased the odds of preeclampsia/eclampsia in pregnant women by 1.56 times compared to women with no family history of preeclampsia/eclampsia. This result is similar to that of a study con-ducted by Boyd et. al. who reported that if a woman has a history of preeclampsia, then her rel-atives will have an increased chance (24–63%) of experiencing the disease [72]. However, although several studies have been conducted to identify the cause for this recurrence, the cause behind the genetic aspect of this disease is still not known [73]. There is strong evidence from different studies that there is a high chance of a recurrence of the disease if the mother has any previous experience or a family history of preeclampsia/eclampsia. Furthermore, it can cause serious and even long-term complications, such as cardiovascular disease. These results imply that during ANC visits a detailed individual and family history of preeclampsia and eclampsia should be taken. This would allow early identification of higher risk women and reduce the complications and serious health outcome.

Pregnant women with a higher BMI were 1.4 times more likely to develop preeclampsia/ eclampsia as compared to women with a lower BMI. This finding is consistent with a study

conducted in Latin America and the Caribbean reported that the risk of preeclampsia/eclampsia increased by nearly three times in overweight and obese women as compared to normal BMI women [74]. Additionally, this is also supported by Bodnar et al. who found obese women have a three to five times higher risk of preeclampsia as compared to normal BMI women [75]. Since obesity and overweight were found to be among the highest risk factors for preeclampsia in the developed world, a gradual increase in the incidence of this condition is expected in developing countries, including SSA countries [76]. Therefore, countries in SSA should pay more attention to this issue and need to include healthy diet as pregnancy preparedness plan and encourage pregnant women to take adequate amount of fruit and vegetable in order to reduce the negative consequences to the mother and foetus.

The presence or absence of maternal pre-existing medical conditions, such as chronic hypertension, diabetes mellitus, renal disease, cardiovascular disease or other diseases are considered a determinant factor for the development of preeclampsia/eclampsia. We found in a meta-analysis of six studies that having chronic hypertension increased the risk of preeclampsia/eclampsia by more than two times as compared to women without chronic hypertension. Several researchers reported about the higher chance of developing preeclampsia and eclampsia in mothers with chronic hypertension. Not only chronic hypertension but also other types of chronic diseases such as diabetes mellitus, renal diseases and others have serious effects on the maternal end organs. Thus, women with one or more complications with such diseases have an increased risk of developing preeclampsia/eclampsia. Therefore, women with chronic hypertension or any other pre-existing medical condition should give due attention during ANC visits and it need preparation even before conceive in order to reduce complication to the mother and the foetus.

In this study, anaemia during pregnancy was found to be strongly associated with the risk of preeclampsia/eclampsia. A WHO study in low and middle-income countries was consistent with our study, in that mothers with severe anaemia have a three-time higher risk of preeclampsia/eclampsia [76]. Similarly, a WHO Global Survey for Maternal and Perinatal Health study found that severe anaemia had a significant association with preeclampsia/eclampsia for nulliparous (OR: 3.55; 95% CI: 2.87, 4.41) and multiparous (OR: 3.94; 95% CI: 3.05, 5.09) women [77]. This indicates that testing and screening for anaemia during ANC visits can reduce the risk of preeclampsia and eclampsia.

Women who have ANC visits are more benefited for better heath to the women and her newborn. In this study, we found that women with no ANC visit nearly threefold the chance of developing preeclampsia /eclampsia. Consistent with these finding, many other studies found the association between lack of ANC visit and increased risk of preeclampsia and eclampsia [78, 79]. If women absence from the regular ANC visits, potential risk factors and danger sign cannot be detected timely. Therefore, early detection and timely management of preeclampsia/eclampsia reduce the disease progress and prevent serious complication to the mother and foetus.

Furthermore, our review highlights that receiving nutritional counselling and dietary diversity were found to be protective factors for preeclampsia and eclampsia. Although we cannot draw conclusions by doing a meta-analysis as the number of studies found per variable was limited, we have a good understanding of how important it is to consider nutritional and related factors to reduce the risk of the disease. While it is important to have regular visits to antenatal clinics, there remains the question of how much detailed information should be taken by the health providers about the pregnant woman's individual and family history of the disease, especially in SSA countries [80]. Therefore, it is highly recommended that healthcare professionals assess each pregnant woman's previous history and family related risks at her booking visit and tailor her antenatal care services.

## Strengths and limitations

This is the first systematic review to quantitatively summarize and conduct a meta-analysis on risk factors for preeclampsia and eclampsia in SSA countries. A rigorous search was conducted from multiple electronic databases. A quality assessment was conducted, and two independent reviewers conducted the screening. Despite this, the review was not without limitations. All studies used in the meta-analysis were cross-sectional and case-control studies and hence do not show causality. Only studies published in English were included, and as a result, papers published in other languages would have been missed. Heterogeneity of the papers is an issue, which means the heterogeneity in our review could have been due to different factors such as tools used in assessing risk factors or the definitions that the researchers used in their studies. For instance, between-study variability on the categorization of maternal age prevented us from pooling the effects of age on preeclampsia and eclampsia. In addition, for some of the risk factors there were not enough studies per risk factor to assess the association with pre-eclampsia and eclampsia and not all SSA countries were represented in this review. Therefore, it need more comprehensive research to conduct in the region.

## Conclusion and recommendations

The factors associated with preeclampsia are multifaceted and interdependent. We found strong evidence that being primiparous, previous preeclampsia/eclampsia, family history of preeclampsia/eclampsia, obesity and overweight, chronic hypertension, anaemia during pregnancy and lack of ANC visits increased the likelihood of preeclampsia/eclampsia. Several factors were found to have mixed, inconclusive or no association with preeclampsia and eclampsia, which are as follows: maternal age, maternal level of education, and alcohol intake during pregnancy. Therefore, interventions need to be designed to address these factors. Although most of the identified factors were reported in different literature, this study conducted for first time in SSA population. Therefore, the findings of this review can be used by SSA countries to develop a screening guideline or checklist for pregnant mothers during ANC visits. In addition, as the evidence evolves, including the risk factors of the disease, health care providers should be trained with up-to-date information and evidence. Therefore, revising the guidelines and service provision material can help health care providers on how to approach pregnant women and identify potential risk factors before serious complications occur. Moreover, it is essential to design effective interventions that focus on primary prevention, mainly by improving ANC services, screening of anaemia, and improving nutritional counselling services.

## Supporting information

**S1 Checklist.**
(DOC)

**S1 File.**
(DOCX)

**S1 Table.**
(DOCX)

## Acknowledgments

The authors would like to thank the University of Newcastle, Australia, for providing us free digital access to the online library. We would also like to thank Mrs Debbie Booth for her assistance in designing the search strategy and help in searching databases.

## Author Contributions

**Conceptualization:** Maereg Wagnew Meazaw, Catherine Chojenta, Deborah Loxton.

**Data curation:** Maereg Wagnew Meazaw, Muluken Dessalegn Muluneh.

**Formal analysis:** Maereg Wagnew Meazaw.

**Investigation:** Maereg Wagnew Meazaw.

**Methodology:** Maereg Wagnew Meazaw, Catherine Chojenta, Deborah Loxton.

**Software:** Maereg Wagnew Meazaw.

**Supervision:** Catherine Chojenta, Deborah Loxton.

**Visualization:** Maereg Wagnew Meazaw.

**Writing – original draft:** Maereg Wagnew Meazaw.

**Writing – review & editing:** Maereg Wagnew Meazaw, Catherine Chojenta, Muluken Dessalegn Muluneh, Deborah Loxton.

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
