## [Decision Letter · Decision Letter 0]

22 Jun 2020

PONE-D-20-13750

Systematic and Meta-analysis of Factors Associated with Preeclampsia and Eclampsia in sub-Saharan Africa

PLOS ONE

Dear Dr. Meazaw,

Thank you for submitting your manuscript to PLOS ONE. After careful consideration, we feel that it has merit but does not fully meet PLOS ONE’s publication criteria as it currently stands. Therefore, we invite you to submit a revised version of the manuscript that addresses the points raised during the review process.

SPECIFIC ACADEMIC EDITOR COMMENTS: Two expert reviewers in the field handled your manuscript. We thank them for their time. Although interest was found in your study, there were several major concerns that arose during review. These comments include: 1) the need to follow journal guidelines on word count, 2) clarification of definitions and statements, 3) more information needs to be provided about this sample of patients, 4) expanding the results about ANC visits, and 5) the conclusions statements needs to be more specific. Please address all of the reviewers' comments in your revised manuscript.

We look forward to receiving your revised manuscript.

Kind regards,

Frank T. Spradley

Academic Editor

PLOS ONE

Reviewers' comments:

Reviewer's Responses to Questions

**Comments to the Author**

1. Is the manuscript technically sound, and do the data support the conclusions?

Reviewer #1: Yes

Reviewer #2: Yes

2. Has the statistical analysis been performed appropriately and rigorously? 

Reviewer #1: Yes

Reviewer #2: Yes

3. Have the authors made all data underlying the findings in their manuscript fully available?

Reviewer #1: Yes

Reviewer #2: Yes

4. Is the manuscript presented in an intelligible fashion and written in standard English?

Reviewer #1: Yes

Reviewer #2: Yes

5. Review Comments to the Author

Reviewer #1: This is a well reviewed paper where the authors systematically examined the factors associated with preeclampsia and eclampsia among pregnant women in South Saharan African countries. Using PRISMA, they looked at articles published between January 2000 and may 2020 and looked at 12 risk factors known to be associated with Preeclampsia/eclampsia. After the systematic and meta analysis they concluded that 7 out of the 12 factors were associated with Preeclampsia/eclampsia. The most importatant risk factors included a history of preeclampsia and eclampsia of the pregnant woman, obesity, not attending ANC, anaemia during pregnancy and any underlying chronic disease. The conclusions support the review data analysed from the systematic and meta analysis. The statistical analysis were appropriate for this type of analysis.

. The abstract word count exceeded the journal's required 300 words likewise the referencing style did not meet the required vancouver style and need to be revised.

Reviewer #2: Introduction:

Reviewers would like the following corrections:

1) According to the American Association of Obstetrics and Gynaecology (ASOG) definition- Authors mean American College of Obstetrics and Gynecology (ACOG).

2) Definition of pre-eclampsia: “preeclampsia is defined as the presence of hypertension (≥140mmHg/90mmHg) after 20 weeks gestation in previously normotensive women on at least two occasions more than four hours apart.”

Authors have used the definition of pre-eclampsia only based on hypertension, which is incorrect per ACOG definition (need proteinuria or in the absence of proteinuria, thrombocytopenia, renal insufficiency, impaired liver function, pulmonary edema, new-onset headache).

3) “This hypertension association with proteinuria +2 and above in a 24 hour collection of urine, arising de novo after the 20th week of gestation in a previously normotensive woman and resolving completely by the 6th postpartum week “- Reviewers suggest revision. Incorrect statement.

Method:

Reviewers would like to know if any of the included studies had patients with pre-eclampsia who went on to develop HELLP syndrome. If this specific subset was not included. Reviewers would like it to be mentioned under exclusion criteria.

Risk factors:

II. Previous Preeclampsia/eclampsia: Reviewers would like to know if any of the studies showed a relationship between history of Early Onset Preeclampsia (EOPE) and risk of pre-eclampsia in future pregnancies.

IV. Maternal Body Mass index

“…Tanzania found that women who were overweight and women who were obese were 1.4...”- Reviewers recommend adding BMI 25-29.9 and >30 next to overweight and obesity, respectively.

MUAC: Reviewers recommend introducing abbreviation in parentheses

V. Maternal pre-existing medical conditions: Reviewers recommend clarifying heading to include only diabetes and chronic hypertension. Was there any association with gestational diabetes with preeclampsia/eclampsia in the studies included?

VI. Anaemia during pregnancy: Reviewer recommends defining cut off for anemia in the studies used (e.g Hb <11). Reviewer also suggests using Iron deficiency anemia instead of anemia during pregnancy, if the studies included are pertaining to Iron deficiency anemia.

VIII: Nutrition and related factors:

“Furthermore, women with hypocalcemia were about eight times more likely to develop preeclampsia compared to those with normal serum calcium”- Reviewers recommend including mean serum calcium level <8.6 in parenthesis.

Minor edit: low plasma vitamin C as compared to “adequated” plasma vitamin C were more likely to develop preeclampsia: Suggests spell check.

IX: Drinking alcohol during pregnancy:

Reviewers recommend quantifying alcohol consumption (e.g glasses/week).

X: ANC visits- recommend Introducing abbreviation in parentheses.

Reviewer recommends summarizing the number of ANC visits with the development of preeclampsia. This may be beneficial for future recommendation and changes in WHO recommended ANC models.

“odds of having preeclampsia/eclampsia as compares to women”: minor edit: suggests spell check.

“ A study conducted in Mozambique showed that women without transport access or had walk to reach health facility for ANC visits were four times more likely to develop eclampsia than women with transport access (OR: 4.31; 95% CI: 2.77, 6.72) (13)”

Reviewer recommend using this study under ANC visits instead of Pregnancy related risk factors. This is a key factor for missed ANC visits in the developing countries.

XI: Pregnancy related risk factors for preeclampsia/eclampsia:

“Whereas another study reported, women had abortion from the same reduced the risk of hypertension in present pregnancy”- Reviewers recommend describing the type of abortion (induced/spontaneous). Do authors mean same father/paternity, thereby implicating and immunological risk factor for preeclampsia?

“In this review, paternity (25, 40), marital status (49), HIV/AIDS (36, 41), place of residence (34, 40, 57), family size (13, 51), maternal sleeping potion (22) , blood type (45, 64), H. pylori seropositivity (16) distance from health facility (52), placental malaria (64) and seasonal variation (19).”- Reviewers would like authors to consider re-writing sentence fragment.

Discussion:

“Not only does a woman having previous preeclampsia/eclampsia have an effect, but also having a family history of preeclampsia/eclampsia has an effect among family members during pregnancy.”- Reviewers would like authors to consider re-writing sentence fragment.

Conclusion:

Reviewers recommend authors to expand on the fact that most of the study findings are well known risk factors of pre-eclampsia/eclampsia in the existing literature. This study confirms what is already known but adds value as it is specific to the SSA population (first time it has been reported in this population).

6. PLOS authors have the option to publish the peer review history of their article (what does this mean?). If published, this will include your full peer review and any attached files.

Reviewer #1: No

Reviewer #2: No

---

## [Author Response · Author response to Decision Letter 0]

3 Jul 2020

Response to reviewers 

Reviewer #1: This is a well reviewed paper where the authors systematically examined the factors associated with preeclampsia and eclampsia among pregnant women in South Saharan African countries. Using PRISMA, they looked at articles published between January 2000 and may 2020 and looked at 12 risk factors known to be associated with Preeclampsia/eclampsia. After the systematic and meta analysis they concluded that 7 out of the 12 factors were associated with Preeclampsia/eclampsia. The most importatant risk factors included a history of preeclampsia and eclampsia of the pregnant woman, obesity, not attending ANC, anaemia during pregnancy and any underlying chronic disease. The conclusions support the review data analysed from the systematic and meta analysis. The statistical analysis were appropriate for this type of analysis. The abstract word count exceeded the journal's required 300 words likewise the referencing style did not meet the required vancouver style and need to be revised.

Response: Thank you for your feedback. As per your feedback on abstract word count, we have now rewrite the abstract to make more concise and within the word count limit (Line 15-36). In addition, we rechecked and revised the referencing style in this version of manuscript.

Reviewer #2: Introduction:

Reviewers would like the following corrections:

1) According to the American Association of Obstetrics and Gynaecology (ASOG) definition- Authors mean American College of Obstetrics and Gynecology (ACOG).

Response: Thank you for your feedback. We have now corrected as per your comments in Line 41-42. 

2) Definition of pre-eclampsia: “preeclampsia is defined as the presence of hypertension (≥140mmHg/90mmHg) after 20 weeks gestation in previously normotensive women on at least two occasions more than four hours apart.”

Authors have used the definition of pre-eclampsia only based on hypertension, which is incorrect per ACOG definition (need proteinuria or in the absence of proteinuria, thrombocytopenia, renal insufficiency, impaired liver function, pulmonary edema, new-onset headache).

Response: we have now revised according to ACOG definition and incorporate the remaining part of definition. (Line 42-45)

3) “This hypertension association with proteinuria +2 and above in a 24 hour collection of urine, arising de novo after the 20th week of gestation in a previously normotensive woman and resolving completely by the 6th postpartum week “- Reviewers suggest revision. Incorrect statement.

Response: The statement has be rewritten as follow in line 46-48

“The hypertension appear after 20 weeks gestation in previously normotensive women on at least two occasions more than four hours apart and resolving completely by the 6th postpartum week”.

Method:

Reviewers would like to know if any of the included studies had patients with pre-eclampsia who went on to develop HELLP syndrome. If this specific subset was not included. Reviewers would like it to be mentioned under exclusion criteria.

Response: In our review, we did not excluded women who developed HELLP syndrome. Based on our review, we found one cross-sectional study that conducted in Cameroon and in that study finding HELLP syndrome were common complications in women with early-onset preeclampsia as compared to late onset preeclampsia. However, based on the aim of our study, we tried to summarize the factors identified for early as well as late onset preeclampsia in general as per the result.

Risk factors:

II. Previous Preeclampsia/eclampsia: Reviewers would like to know if any of the studies showed a relationship between history of Early Onset Preeclampsia (EOPE) and risk of pre-eclampsia in future pregnancies.

Response: In this review, we found one study that compared early and late onset preeclampsia with maternal and fetal complication. In that study, predictors of the disease as well as short- term maternal and fetal outcomes were studied. However, this study does not identified whether the history of preeclampsia early or late onset preeclampsia. 

IV. Maternal Body Mass index

“…Tanzania found that women who were overweight and women who were obese were 1.4...”- Reviewers recommend adding BMI 25-29.9 and >30 next to overweight and obesity, respectively.

MUAC: Reviewers recommend introducing abbreviation in parentheses

Response: we have now incorporate both comments in Line 245 and 250, respectively. 

V. Maternal pre-existing medical conditions: Reviewers recommend clarifying heading to include only diabetes and chronic hypertension. Was there any association with gestational diabetes with preeclampsia/eclampsia in the studies included?

Response: The heading revised as per the comment (Line 268). In addition, based on our review, none of the included studies mentioned about gestational diabetes in their study. 

VI. Anaemia during pregnancy: Reviewer recommends defining cut off for anemia in the studies used (e.g Hb <11). Reviewer also suggests using Iron deficiency anemia instead of anemia during pregnancy, if the studies included are pertaining to Iron deficiency anemia.

Response: we have now incorporated the information for the cut off for anaemia diagnosis with the following statement (Line 284-286)

“In this review, four of the studies mentioned the cut off point for presence of anaemia as hemoglobin level <11 g/dL (27, 46, 48, 53), whereas one study report hemoglobin level <12 g/dL as a diagnosis criteria for anaemia (24)” In addition, none of included studies mentioned about the type of anaemia.

VIII: Nutrition and related factors:

“Furthermore, women with hypocalcemia were about eight times more likely to develop preeclampsia compared to those with normal serum calcium”- Reviewers recommend including mean serum calcium level <8.6 in parenthesis.

Minor edit: low plasma vitamin C as compared to “adequate” plasma vitamin C were more likely to develop preeclampsia: Suggests spell check.

Response: Both information incorporated in this version of manuscript in line 340 and 346.

IX: Drinking alcohol during pregnancy:

Reviewers recommend quantifying alcohol consumption (e.g. glasses/week).

Response: In our review, none of the included studies reported the level of alcohol consumption. Based on the feedback, we have now included this information in the revised manuscript as follow (line 353-354).

“Based on this review, none of the included studies reported the quantifying level of alcohol consumption in their reports.” 

X: ANC visits- recommend Introducing abbreviation in parentheses.

Reviewer recommends summarizing the number of ANC visits with the development of preeclampsia. This may be beneficial for future recommendation and changes in WHO recommended ANC models.

“odds of having preeclampsia/eclampsia as compares to women”: minor edit: suggests spell check.

“ A study conducted in Mozambique showed that women without transport access or had walk to reach health facility for ANC visits were four times more likely to develop eclampsia than women with transport access (OR: 4.31; 95% CI: 2.77, 6.72) (13)”

Reviewer recommend using this study under ANC visits instead of Pregnancy related risk factors. This is a key factor for missed ANC visits in the developing countries.

Response: Based on our review, women with no ANC visits found a higher risk developing preeclampsia and eclampsia. We have incorporated this information in discussion section (line 505-5011)

“Women who have ANC visits are more benefited for better heath to the women and her newborn. In this study, we found that women with no ANC visit nearly threefold the chance of developing preeclampsia /eclampsia. Consistent with these finding, many other studies found the association between lack of ANC visit and increased risk of preeclampsia and eclampsia (78, 79). If women absence from the regular ANC visits, potential risk factors and danger sign cannot be detected timely. Therefore, early detection and timely management of preeclampsia/eclampsia reduce the disease progress and prevent serious complication to the mother and foetus”

In addition, we included the abbreviation and rewritten the sentence mentioned for spell check. In addition, we moved mentioned study to “ANC visit” section. (Line 369-370)

XI: Pregnancy related risk factors for preeclampsia/eclampsia:

“Whereas another study reported, women had abortion from the same reduced the risk of hypertension in present pregnancy”- Reviewers recommend describing the type of abortion (induced/spontaneous). Do authors mean same father/paternity, thereby implicating and immunological risk factor for preeclampsia? 

“In this review, paternity (25, 40), marital status (49), HIV/AIDS (36, 41), place of residence (34, 40, 57), family size (13, 51), maternal sleeping potion (22), blood type (45, 64), H. pylori seropositivity (16) distance from health facility (52), placental malaria (64) and seasonal variation (19).”- Reviewers would like authors to consider re-writing sentence fragment.

Response: This specific study excluded women with spontaneous abortion and the cohort included women with induced abortion. We have now incorporated the type of abortion and paternity, and rewritten the fragmented sentence as follow (Line 413-416). 

“Whereas another study reported, women had the same paternity induced abortion nearly 50% reduced the risk of hypertension in present pregnancy as compared to women without same paternity induced abortion (OR: 0.48; 95% CI: 0.31,0.73), and thus this finding support hypothesis the role of immunological factors in the pathophysiological mechanism preeclampsia”

Similarly, we rewrite the second comment on the following statement (line 424-427)

“In this review, the following factors were identified as a risk factors for preeclampsia and eclampsia: paternity (25, 40), marital status (49), HIV/AIDS (36, 41), place of residence (34, 40, 57), family size (13, 51), maternal sleeping position (22) , blood type (45, 64), H. pylori seropositivity (16), distance from health facility (52), placental malaria (64) and seasonal variation (19).”

Discussion:

“Not only does a woman having previous preeclampsia/eclampsia have an effect, but also having a family history of preeclampsia/eclampsia has an effect among family members during pregnancy.”- Reviewers would like authors to consider re-writing sentence fragment.

Response: We have now rewritten the fragmented statement as follow (line 460-463)

“Not only woman having previous preeclampsia/eclampsia have an increased risk of preeclampsia/eclampsia, but also having a family history of preeclampsia/eclampsia has positive association with present pregnancy.”

Conclusion:

Reviewers recommend authors to expand on the fact that most of the study findings are well known risk factors of pre-eclampsia/eclampsia in the existing literature. This study confirms what is already known but adds value as it is specific to the SSA population (first time it has been reported in this population).

Response: The comment is accepted and we have included the information accordingly (line 548-549).

“Although most of the identified factors were reported in different literature, this study conducted for first time in SSA population. Therefore, the findings of this review can be used by SSA countries to develop a screening guideline or checklist for pregnant mothers during ANC visits.”

---

## [Decision Letter · Decision Letter 1]

23 Jul 2020

PONE-D-20-13750R1

Systematic and Meta-analysis of Factors Associated with Preeclampsia and Eclampsia in sub-Saharan Africa

PLOS ONE

Dear Dr. Meazaw,

Thank you for submitting your manuscript to PLOS ONE. After careful consideration, we feel that it has merit but does not fully meet PLOS ONE’s publication criteria as it currently stands. Therefore, we invite you to submit a revised version of the manuscript that addresses the points raised during the review process.

SPECIFIC ACADEMIC EDITOR COMMENT: There is a minor revision requested.

We look forward to receiving your revised manuscript.

Kind regards,

Frank T. Spradley

Academic Editor

PLOS ONE

Reviewers' comments:

Reviewer's Responses to Questions

**Comments to the Author**

1. If the authors have adequately addressed your comments raised in a previous round of review and you feel that this manuscript is now acceptable for publication, you may indicate that here to bypass the “Comments to the Author” section, enter your conflict of interest statement in the “Confidential to Editor” section, and submit your "Accept" recommendation.

Reviewer #1: All comments have been addressed

Reviewer #2: All comments have been addressed

2. Is the manuscript technically sound, and do the data support the conclusions?

Reviewer #1: (No Response)

Reviewer #2: Yes

3. Has the statistical analysis been performed appropriately and rigorously? 

Reviewer #1: (No Response)

Reviewer #2: Yes

4. Have the authors made all data underlying the findings in their manuscript fully available?

Reviewer #1: (No Response)

Reviewer #2: Yes

5. Is the manuscript presented in an intelligible fashion and written in standard English?

Reviewer #1: (No Response)

Reviewer #2: Yes

6. Review Comments to the Author

Reviewer #1: (No Response)

Reviewer #2: Excellent work! Thank for your corrections. Only minor edit recommended would be:

(line 460-463)

“Not only woman having previous preeclampsia/eclampsia have an increased risk of preeclampsia/eclampsia, but also having a family history of preeclampsia/eclampsia has positive association with present pregnancy.”

Women with obstetric and family histories of preeclampsia/eclampsia were noted to be at an increased risk for development of preeclampsia/eclampsia in the subsequent pregnancies.

7. PLOS authors have the option to publish the peer review history of their article (what does this mean?). If published, this will include your full peer review and any attached files.

Reviewer #1: No

Reviewer #2: No

---

## [Author Response · Author response to Decision Letter 1]

27 Jul 2020

Response to reviewers

Reviewer #: Thank you once again for your constructive feedback and comments.

Reviewer #2: Excellent work! Thank for your corrections. Only minor edit recommended would be:

(line 460-463)

“Not only woman having previous preeclampsia/eclampsia have an increased risk of preeclampsia/eclampsia, but also having a family history of preeclampsia/eclampsia has positive association with present pregnancy.”

Response: Thank you for your constructive comment that will make the manuscript more readable. We have rewritten the sentence as per your guide and advice. (Line 444-445)

“Women with family histories of preeclampsia/eclampsia were noted to be at an increased risk for development of preeclampsia/eclampsia in the subsequent pregnancies.”

---

## [Editor Report · Decision Letter 2]

30 Jul 2020

Systematic and Meta-analysis of Factors Associated with Preeclampsia and Eclampsia in sub-Saharan Africa

PONE-D-20-13750R2

Dear Dr. Meazaw,

We’re pleased to inform you that your manuscript has been judged scientifically suitable for publication and will be formally accepted for publication once it meets all outstanding technical requirements.

Kind regards,

Frank T. Spradley

Academic Editor

PLOS ONE

---

## [Editor Report · Acceptance letter]

3 Aug 2020

PONE-D-20-13750R2 

Systematic and Meta-analysis of Factors Associated with Preeclampsia and Eclampsia in sub-Saharan Africa 

Dear Dr. Meazaw:

I'm pleased to inform you that your manuscript has been deemed suitable for publication in PLOS ONE. Congratulations! Your manuscript is now with our production department. 

Kind regards, 

on behalf of

Dr. Frank T. Spradley 

Academic Editor

PLOS ONE